# Recommendations for the Clinical Approach to Immune Thrombocytopenia: Spanish ITP Working Group (GEPTI)

**DOI:** 10.3390/jcm12206422

**Published:** 2023-10-10

**Authors:** María Eva Mingot-Castellano, Mariana Canaro Hirnyk, Blanca Sánchez-González, María Teresa Álvarez-Román, Abelardo Bárez-García, Ángel Bernardo-Gutiérrez, Silvia Bernat-Pablo, Estefanía Bolaños-Calderón, Nora Butta-Coll, Gonzalo Caballero-Navarro, Isabel Socorro Caparrós-Miranda, Laura Entrena-Ureña, Luis Fernando Fernández-Fuertes, Luis Javier García-Frade, María del Carmen Gómez del Castillo, Tomás José González-López, Carlos Grande-García, José María Guinea de Castro, Isidro Jarque-Ramos, Reyes Jiménez-Bárcenas, Elsa López-Ansoar, Daniel Martínez-Carballeira, Violeta Martínez-Robles, Emilio Monteagudo-Montesinos, José Antonio Páramo-Fernández, María del Mar Perera-Álvarez, Inmaculada Soto-Ortega, David Valcárcel-Ferreiras, Cristina Pascual-Izquierdo

**Affiliations:** 1Hematology Department, Hospital Universitario Virgen del Rocío, Instituto de Biomedicina de Sevilla, 41013 Sevilla, Spain; 2Hematology Department, Hospital Son Espases, Palma de Mallorca, 07210 Palma, Spain; mariana.canaro@ssib.es; 3Hematology Department, Hospital del Mar, 08003 Barcelona, Spain; bsanchezgonzalez@psmar.cat; 4Hematology Department, Hospital Universitario La Paz-IdiPAZ, Universidad Autónoma de Madrid, 28046 Madrid, Spain; maroman@salud.madrid.org; 5Hematology Department, Hospital de Ávila, 05004 Ávila, Spain; abarez@saludcastillayleon.es; 6Hematology Department, Hospital Central de Asturias, 33011 Oviedo, Spain; angel.bernardo@sespa.es (Á.B.-G.); daniel_mc@hotmail.es (D.M.-C.); inmaculada.soto@sespa.es (I.S.-O.); 7Hematology Department, Hospital Universitario de la Plana, 12540 Villarreal, Spain; bernat_sil@gva.es; 8Hematology Department, Hospital Clínico San Carlos, IML, IdISSC, 28040 Madrid, Spain; estefaniabolanosc@gmail.com; 9Hematology Department, Instituto de Investigación Hospital Universitario La Paz (IdiPAZ), 28046 Madrid, Spain; nora.butta@salud.madrid.org; 10Hematology Department, Hospital Universitario Miguel Servet, 50009 Zaragoza, Spain; gcaballeron@salud.aragon.es; 11Hematology Department, Hospital Universitario Virgen de la Victoria, 29010 Málaga, Spain; isabels.caparros.sspa@juntadeandalucia.es; 12Hematology Department, Hospital Universitario Virgen de las Nieves, 18014 Granada, Spain; laura.entrena.sspa@juntadeandalucia.es; 13Hematology Department, Complejo Hospitalario Universitario Insular Materno-Infantil, 35016 Las Palmas de Gran Canaria, Spain; fferfue@gobiernodecanarias.org; 14Hematology Department, Hospital Universitario Río Hortega, Gerencia Regional de Salud de Castilla y León, 47012 Valladolid, Spain; lgarciafr@saludcastillayleon.es; 15Hematology Department, Hospital General de la Coruña, 15006 A Coruña, Spain; ma.del.carmen.gomez.del.castillo.solano@sergas.es; 16Hematology Department, Hospital Universitario de Burgos, 09006 Burgos, Spain; tjgonzalez@saludcastillayleon.es; 17Hematology Department, Clínica Universidad de Navarra, 28022 Madrid, Spain; carlos.grande@salud.madrid.org; 18Hematology Department, Hospital Universitario de Araba, 01009 Vitoria-Gasteiz, Spain; josemaria.guineadecastro@osakidetza.eus; 19Hematology Department, Hospital Universitario y Politécnico La Fe, 46026 Valencia, Spain; jarque_isi@gva.es; 20Hematology Department, Hospital Serranía de Ronda, 29400 Ronda, Spain; reyes.jimenez.sspa@juntadeandalucia.es; 21Hematology Department, Complejo Hospitalario Universitario de Vigo, 36312 Vigo, Spain; elsa.lopez.ansoar@sergas.es; 22Hematology Department, Complejo Asistencial Universitario de León, 24008 León, Spain; viomartinez@saludcastillayleon.es; 23Pediatric Department, Hospital Universitario y Politécnico La Fe, 46026 Valencia, Spain; monteagudo_emi@gva.es; 24Hematology Department, Clínica Universidad de Navarra, 31008 Pamplona, Spain; japaramo@unav.es; 25Hematology Department, Hospital Universitario de Gran Canaria Doctor Negrín, 35010 Las Palmas de Gran Canaria, Spain; mperalvm@gobiernodecanarias.org; 26Hematology Department, Vall d’Hebron Instituto de Oncología (VHIO), Universitat Autònoma de Barcelona, 08035 Barcelona, Spain; dvalcarcel@vhio.net; 27Hematology Department, Hospital General Universitario Gregorio Marañón (HGUGM) Madrid, Instituto de Investigación Gregorio Marañón, 28007 Madrid, Spain; cpascuali@salud.madrid.org; 28Spanish Immune Thrombocytopenia Group, 28040 Madrid, Spain

**Keywords:** primary immune thrombocytopenia, glucocorticoids, intravenous immunoglobulins, fostamatinib, rituximab, recommendations

## Abstract

Primary immune thrombocytopenia (ITP) is a complex autoimmune disease whose hallmark is a deregulation of cellular and humoral immunity leading to increased destruction and reduced production of platelets. The heterogeneity of presentation and clinical course hampers personalized approaches for diagnosis and management. In 2021, the Spanish ITP Group (GEPTI) of the Spanish Society of Hematology and Hemotherapy (SEHH) updated a consensus document that had been launched in 2011. The updated guidelines have been the reference for the diagnosis and management of primary ITP in Spain ever since. Nevertheless, the emergence of new tools and strategies makes it advisable to review them again. For this reason, we have updated the main recommendations appropriately. Our aim is to provide a practical tool to facilitate the integral management of all aspects of primary ITP management.

## 1. Introduction

### 1.1. Definition and Epidemiology

Primary ITP is defined as a platelet count < 100 × 10^9^/L whose cause has not been identified. The concept of “secondary ITP” arises from scenarios where low platelet counts are subsequent to diagnosed diseases known to cause immune destruction of platelets. The incidence of primary ITP is two to four cases per 100,000 individuals per year, in both adults and children. The prevalence is higher in adults (10 per 100,000 vs. 5 per 100,000 individuals in children) because the rate of chronicity is greater in this population [1].

### 1.2. Diagnosis

Despite the important advances achieved in the therapeutic field in the last decade, there has been little progress in the diagnosis of the disease, and reliable markers or confirmatory tests are lacking. Diagnosis continues to be essentially clinical and based on excluding other causes of thrombocytopenia. When a patient is suspected of primary ITP, once the clinical history, physical examination, peripheral blood smear, immunoglobulin level and viral serology have allowed us to rule out other processes, the general rule consists of limiting tests to a minimum, since most of them will not be very informative. The paradigmatic laboratory finding is isolated thrombocytopenia, and careful examination of the peripheral blood smear is mandatory. In primary ITP patients, platelets are usually large and granular, with elevated mean platelet volumes and immature fractions. The systematic analysis of bone marrow is not recommended, except in the event that treatment response is inadequate, or when other abnormalities in the peripheral blood smear or the clinical presentation lead us to suspect other disorders. These limitations may result in an incorrect diagnosis in one out of seven patients identified as having primary ITP [2]. Further measures to overcome this challenge are required.

### 1.3. Etiology

There are many pathophysiological mechanisms that play a causal role in primary ITP, which explains the heterogeneity of the disease. Nevertheless, the consequences are basically of two types. On the one hand, there is an increase in platelet destruction, which is mainly, but not solely, caused by the onset of autoantibodies able to opsonize the cell surface for the subsequent complement- or phagocyte-mediated cell killing. The increased rate of platelet desialylation, which accelerates liver clearance, and a greater apoptosis rate also contribute to the low counts. On the other hand, platelet turnover decreases because of a lower rate of cell production. This is owing to autoantibodies targeting the thrombopoietin (TPO) receptor and preventing TPO from stimulating the proliferation and differentiation of megakaryocytes, as well as an apoptotic imbalance concerning these cells [3].

The autoantibodies identified in patients with primary ITP can bind a large variety of targets that are frequently located at the platelet surface, for instance, glycoproteins (GP) GPIIb/IIIa or GPIb/IX. Bleeding manifestations are not the only symptoms of primary ITP. Fatigue is also frequently found, and some patients are predisposed to experience thromboembolic events, infection or other autoimmune diseases. Primary ITP shows a self-limiting course in the majority of children and in one-third of adults. According to the standardized terminology [4], the disease can be defined according to the time elapsed since diagnosis: (i) “newly diagnosed primary ITP” encompasses all cases at diagnosis; (ii) “persistent primary ITP” refers to the period lasting between 3 and 12 months from diagnosis and (iii) “chronic primary ITP” is the term reserved for patients with primary ITP lasting for more than 12 months.

### 1.4. Outcome

Somehow paradoxically, patients with primary ITP are at an increased risk of thrombosis. Quality of life (QoL) of primary ITP patients is reduced to a similar extent to that seen in patients with other chronic diseases such as cancer, arthritis or diabetes mellitus [5]. The proper treatment of primary ITP not only has to pursue the recovery of platelet counts and cessation of bleeding. Minimizing the impact of the disease on the patient’s QoL is highly advisable. Table 1 summarizes the most relevant topics regarding pathophysiology and diagnosis of primary ITP, and Table 2 compiles a list of definitions and concepts that have reached consensus and should be well known [4,6].

### 1.5. Aim of This Review

In the last 3 years, new drugs have been developed, and some treatment recommendations have been critically reviewed. Thus, our aim is to offer the reader an updated practical tool to enable the integral management of all aspects of primary ITP. The added value this review provides beyond the other available literature consists of introducing prophylaxis against infections and bone damage when using steroids in daily clinical practice; acknowledging the development of new drugs to be used as second-line options; providing recommendations for tapering of TPO-receptor agonists (TPO-RAs); providing new information regarding the management of pregnant patients; providing the most updated data regarding potential associations between SARS-CoV-2 infection and ITP and finally, considering therapeutic options such as the combination of TPO-RAs with other immunosuppressants as potential second- and third-line treatments.

## 2. Methods

The AGREE methodology was followed to compile these recommendations. The PICO (Population, Intervention, Comparison, and Outcome) framework was used to select the questions to be addressed regarding ITP management. These were as follows: first-line, second-line and multirefractory ITP treatment; follow-up of patients with primary ITP; primary ITP in selected patient populations; secondary ITP; primary ITP and thrombosis and ITP and COVID-19.

Each author was assigned one topic to perform a comprehensive literature search, especially focusing on the last 5 years and guided by relevant MeSH terms. In order to establish the recommendations, the reliability of the compiled information was evaluated, according to strengths and limitations. Finally, peer reviews of each topic were performed until a general consensus was reached.

## 3. First-Line, Second-Line and Multirefractory ITP Treatment

Table 3 summarizes the main issues to consider regarding first and second-line treatment, as well as those scenarios of recurrent refractoriness, and sets out the response rates expected for each therapeutic strategy [1,6,7,8,9].

### 3.1. First-Line Treatment

The aim of the treatment is to achieve hemorrhage cessation and prevent future bleeding events. Treatment must be started in newly diagnosed adult patients with active bleeding or when they present with platelet counts < 20 × 10^9^/L (<30 × 10^9^/L when they are >65 years old or present with hemorrhage risk factors). These criteria are not necessarily to be applied for a second treatment since the patient’s opinion is particularly important in such a situation. First-line treatment has not evolved dramatically. Table 4 shows the main therapeutic options with their expected response rates. Glucocorticoids remain the cornerstone, although treatment duration has been reduced to minimize side effects. The initial dose of prednisone (0.5–1 mg/kg/day, not exceeding 80 mg daily) should not be maintained beyond 3 weeks (two weeks in the event of no response). The dose must be progressively reduced, and treatment must be terminated no later than 8 weeks from the start. Dexamethasone, at no more than three to four cycles consisting of 40 mg/day for 4 days each 2–4 weeks, is a validated alternative. Although the recovery of platelet counts is faster with the latter, long-term response rates are not different for the two therapies [10]. Intravenous (i.v.) immunoglobulins (IVIg) are recommended for patients with active bleeding or when steroids are contraindicated. The more widely used regimens are 1 g/kg administered 1 or 2 days, or 0.4 g/kg administered 3–5 days in patients >65 years old. Nevertheless, alternative patterns have been suggested, such as a single dose of 0.2–0.4/kg, which could be repeated again 3 days afterward in the event of no response. This last strategy has been shown to be effective and, furthermore, appears to be more sustainable [11].

Other therapeutic options have been suggested, although the experience reported so far does not allow us to consider them as being on the same level as those mentioned above. Combined therapies have been tested: dexamethasone with rituximab may provide a better long-term response in adult ITP patients, with no major risk of adverse effects [12]; the combination of mycophenolate mofetil with a glucocorticoid resulted in improved response and a lower risk of refractory or relapsed ITP [13]; and other options whose results have not been made available yet, such as the combination of TPO-RAs with steroids [14,15].

Anti-D immunoglobulin is another tool that may be considered among the first-line options. However, this treatment is not available worldwide. Doses (single, i.v.) are in the range of 50–75 μg/kg, and close monitoring is recommended in Rh(D) positive patients to prevent secondary hemolytic anemia [16,17]. Finally, the promising results obtained with vitamin D supplementation in children with persistent and chronic ITP suggest that vitamin D may play an immunomodulatory role in ITP and, therefore, may be a therapeutic target [18].

In the event of severe hemorrhage, high-dose methylprednisolone and platelet transfusion are recommended in addition to IVIg, and the use of TPO-RAs can be considered. Among these, romiplostim at 5–10 μg/kg is the most frequently reported option [19], although there is no reason to think that eltrombopag at a daily dose of 75 mg will not be effective. The efficacy of treatments to stop severe hemorrhage has to be assessed according to blood cessation rather than platelet count recovery [20]. Finally, the combination of steroids with either rituximab, TPO-RAs or immunosuppressants is not recommended outside of clinical trials [13,21,22,23].

Prophylaxis with trimetropin–sulfametoxazol at 80 mg/400 mg twice a day two to three times a week to prevent infection by *Pneumocystis carinii* must be administered in the following cases: patients on steroid treatment lasting >4 weeks at daily doses >30 mg; patients on prednisone > 8 weeks at 15–30/mg/day; patients combining 15–30/mg/day prednisone with cyclosporine; and patients with prednisone at >10 mg/day and meeting ≥ two of the following criteria: age >65 years, pulmonary disease and concomitant use of another immunosuppressant. Prophylaxis against herpes virus with acyclovir at 400 mg/day is advisable for patients >60 years old, patients on prednisone at daily doses >7.5 mg or patients with a history of infection with this pathogen.

Prophylaxis with entecavir at 0.5 mg/day is recommended for those patients with antibodies against VHBc and a positive test for hepatitis B virus (HBV) antigen, who are on treatment with prednisone either at >10 mg/day during ≥8 weeks, or at >20 mg/day during >4 weeks. In the event that the test for antibodies against VHBc was positive but that for HBV antigen was negative, the patient should be periodically monitored [24].

Prophylaxis to prevent osteoporosis with calcium and vitamin D (colecalciferol at a weekly dose of 2800 IU) is recommended for postmenopausal women, >50 years old (y.o.) male patients who have been on steroid treatment for >3 months, and in those patients who have a T-score of bone mineral density (BMD) < −1.5 and are being treated or are to be treated with steroids at doses >2.5 mg/day for >3 months. This prophylaxis should be applied to premenopausal women and <50 y.o. male patients only in the event of a history of previous fractures or when the T-score is <−1.5 and treatment with steroids at doses >5 mg/day for >3 months is being administered or planned [25].

### 3.2. Second-Line Treatment

Although initial response rates to glucocorticoids are high, many adult patients will relapse (Table 3). In these cases, re-exposing patients to these treatments is not suitable unless it is justified by an emergency situation. Personalizing therapy becomes paramount when choosing a second-line treatment option. Thus, each patient’s comorbidities will notably influence the therapeutic decision. The findings observed in studies with TPO-RAs, fostamatinib and rituximab suggest that the first two are the more effective and less toxic therapeutic options to be used as second-line treatment of primary ITP [26,27,28,29,30,31,32,33,34,35,36]. Randomized studies to compare them directly have not been reported so far.

TPO-RAs induce platelet production and have an excellent efficacy/safety profile. We recommend using any of the commercially available TPO-RAs as the first option for second-line treatment, although experience with eltrombopag and romiplostin is longer than that reported with avatrombopag so far. Patients will actively participate in decision-making, and the choice will also be influenced by their priorities and lifestyle. Responses have been reported in >80% of cases with these agents [26,27], and cross-resistance has not been observed [28]. Furthermore, according to several real-world studies [37,38,39,40], a number of patients ranging between 30 and 50% will be able to suspend other treatments, even TPO-RAs themselves, without leading to a new drop in platelet counts. In the absence of a response to a TPO-RA, switching to another one or to fostamatinib is recommended. If TPO-RA refractoriness is definitely confirmed, the use of other immunosuppressants such as rituximab, mycophenolate mofetil, azathioprine or low-dose steroids can be considered [29].

Fostamatinib is another option for second-line treatment in ITP. Fostamatinib is a spleen tyrosine kinase (SYK) inhibitor able to reduce the anti-platelet activity of phagocytes. This agent achieves rapid, long-lasting platelet count increases in 40–45% of those patients refractory to previous treatments [30]. Although the response rate is lower when considering heavily pre-treated patients, the efficacy of fostamatinib goes up to 75% when chosen as the first option for second-line treatment [31]. Furthermore, some studies have shown encouraging results regarding long-term efficacy, with roughly 65% of patients showing responses with a median duration of >28 months and median platelet counts of 63–89 × 10^9^/L and 89 × 10^9^/L [32,33]. We recommend fostamatinib as a second-line treatment even before TPO-RA in patients with high thromboembolic risk. The remarkably low incidence of thromboembolic events, together with the absence of platelet peaks associated with fostamatinib, make it particularly suitable for use by primary ITP patients presenting with either arterial or venous thrombosis or a history of previous thromboembolic events, regardless of their severity [33].

Rituximab is a monoclonal antibody targeting the B-cell surface receptor CD20. The interaction induces B-cell depletion and the subsequent decrease in antibody generation. This agent is the second option for second-line treatment. The experience with rituximab in primary ITP patients is extensive even though it has no specific approval to treat this disease [34]. Overall responses have been reported in 60–80% of patients, although the proportion of those achieving long-lasting responses after >3–5 years drops to 20–30% [35,36]. Rituximab has been shown to be particularly effective in those patients positive for antinuclear antibodies (ANA) since ANA positivity was significantly related to a better cumulative incidence of overall response [41]. The standard and more widely used regimen consists of four doses of 375 mg/m^2^ administered over 4 consecutive weeks. Nevertheless, the low-dose regimen, which uses four doses of 100 mg/m^2^ instead of 375 mg/m^2^, makes it possible to save costs and is associated with fewer adverse events, while showing a similar efficacy [42]. There is a third regimen consisting of 1 g/day doses administered on days 1 and 15, whose efficacy is similar to the previous ones [43]. In any case, vaccination against encapsulated bacteria is required before starting rituximab. Active or latent HBV infection has also to be discarded, and, when applicable, treated. Finally, it must be borne in mind that progressive multifocal leukoencephalopathy has been classified as a complication of rituximab treatment, although it occurs very rarely [36].

Another second-line option is splenectomy. Its main advantage is its high efficacy associated with a low cost (Table 3) [44]. There are also major limitations, such as the increased risk of thromboembolism or severe infection [45]. Nevertheless, this may be a matter of controversy, since there are also studies whose results do not support a worryingly increased risk of infection or thrombosis [46]. In any case, the benefit:risk ratio must be carefully assessed. Since the current scenario offers several second-line safe and effective pharmacologic options (with others in the pipeline), GEPTI recommends limiting splenectomy to highly selected patients according to their comorbidities, lifestyle and priorities, as well as delaying the procedure in the hope that the patient may achieve a suboptimal response at least to one of the second-line therapies. More specifically, splenectomy should be considered in those cases diagnosed at least 12 months previously after failure of other therapeutic schemes, who are not willing to receive medication on a chronic basis. In any case, splenectomy must not be performed within the first 12 months after diagnosis. In the event that splenectomy is finally the chosen option, the laparoscopic procedure is preferred, and an appropriate preoperative vaccine pattern and postoperative thromboembolic prophylaxis must be observed [1,47].

### 3.3. Vaccination Prior to Splenectomy

After the procedure, those patients who have not been properly immunized are at a risk of severe bacterial infection that is 50-fold higher than that of non-splenectomized patients. The causal agents are *Streptococcus pneumoniae*, *Haemophilus influenzae* and *Neisseria meningitidis* in 50–90%, 5–15% and 5–15% of cases, respectively [48]. Vaccination reduces the risk, but this does not disappear completely. Vaccines must be administered at least 2 weeks before the procedure. In the event that the severity of the situation prompts immediate surgical intervention, vaccination will be initiated as soon as possible, within the first 2 weeks after the procedure.

Finally, these patients must also be vaccinated against influenza each year, and the serogroup B meningococcal vaccine has to be considered for those younger than 25 years [49].

### 3.4. Treatment of Multirefractory Patients

Multirefractory primary ITP is a severe condition that can be experienced by up to 20% of patients. The term “refractoriness” has been controversial. It has been recently defined as the total loss of response to one or more treatments, including rituximab and TPO-RA [50]. In these cases, reconsidering primary ITP diagnosis is advisable, and bone marrow examination is indicated. As far as the therapeutic attitude is concerned, eradication of *Helicobacter pylori*, a gram-negative bacterium that can be detected in the digestive tract of more than half of the total population, can be proposed, since it has been associated with primary ITP in several studies [51]. On the other hand, treatments consisting of combinations of agents able to induce platelet generation and prevent platelet destruction have been applied. Patient rescues subsequent to administration of steroids concomitantly with rituximab or TPO-RA, as well as subsequent to rituximab and TPO-RA combination, have been described [52,53,54,55,56].

The use of immunosuppressants such as azathioprine, cyclosporine or mycophenolate mofetil, immunomodulators such as danazol or dapsone, or cytostatic agents such as cyclophosphamide or vinca alkaloids (vincristine, vinblastine) can also be envisaged [52,57,58,59,60,61,62]. Nevertheless, these agents may induce side effects that should prompt a careful examination of the benefit:risk ratio. Some patients may present without active bleeding and with no limitation in their QoL, and thus do not require pharmacologic support while their condition persists. Furthermore, there are no reliable studies either providing support for their use or comparing their efficacy. The Appendix A provides details regarding dose, expected response and side effects associated with the treatments addressed in this section.

## 4. Follow-Up of Patients with Primary ITP—Scenarios and Recommendations

The fact that the diagnosis of primary ITP is performed by exclusion may lead to situations where the definitive diagnosis has not been made before the initiation of treatment. Patients should be closely followed up by experienced practitioners, with the aim to rule out other diseases responsible for the symptoms attributed to primary ITP, and to control the subsequent onset of other disorders, especially when patients have persistent or chronic primary ITP, or are elderly. Furthermore, follow-up is required for the early identification of thrombocytopenia-derived complications and side effects of treatments. The scenarios that can be most frequently found throughout the follow-up period are described below.

### 4.1. Hospitalization

The hospital admittance criteria for primary ITP patients are as follows [1]:Grade 2 hemorrhage according to the World Health Organization (WHO), and platelets < 30 × 10^9^/L.Grade ≥3 hemorrhage (requires red blood cell transfusion), regardless of platelet counts.Adults who are newly diagnosed with primary ITP and present with platelet counts < 20 × 10^9^/L, even if they are asymptomatic or present with minor mucocutaneous hemorrhage. This decision is supported by the following arguments: possible uncertainty regarding diagnosis; the requirement to monitor platelet count evolution; possible bleeding complications; and the need to guarantee that treatment is administered correctly.The following patient profiles could also benefit from hospitalization:
○Those refractory to treatment.○Those whose diagnosis is not reliable enough.○Those presenting with relevant comorbidities.○Those using concomitant medication are associated with high hemorrhagic risk.○Those presenting with significant mucosal bleeding.○Those either with low social support, living far away from the hospital or whose follow-up cannot be guaranteed.



Those adult patients with newly diagnosed primary ITP with platelet counts > 20 × 10^9^/L who either are asymptomatic or present with minor mucocutaneous bleeding are recommended to receive ambulatory treatment instead of hospitalization. Table 4 summarizes the guidelines to follow with the different profiles of primary ITP patients who are not hospitalized [1,6,8,9,63].

### 4.2. Follow-Up of Diseases Frequently Associated with Primary ITP

During follow-up, close monitoring for early detection of diseases classically overrepresented in primary ITP patients is advisable. The prevalence of diabetes, renal failure, hypertension, vascular disease and thyroid disease is 2–2.5-fold higher than that of the normal population, the prevalence of other autoimmune diseases is 5-fold higher and that of hematological malignancies is up to 6–20-fold higher.

### 4.3. Surgery

The optimal platelet count target to avoid surgery-associated risk is still controversial. As a general rule, it is accepted that presurgical treatment is required when platelet counts are <50 × 10^9^/L, while it would not be needed with counts > 100 × 10^9^/L [64]. Nevertheless, these values not only are merely indicative, but they are not directly applicable to primary ITP either, since bleeding manifestations are less frequent in patients with this condition than in patients with other thrombocytopenias [65]. For minor procedures with a standard bleeding risk, platelet counts > 50 × 10^9^/L are recommended, which should increase to >70–100 × 10^9^/L to undergo major surgery or procedures on the central nervous system.

In emergency situations, the approach must be the same as that followed in the scenario of severe bleeding, i.e., one or more of these actions should be taken: administration of IVIg; administration of corticosteroids, preferably dexamethasone to take advantage of its rapid-acting profile; and platelet transfusion, ideally after the aforementioned measures have been applied. TPO-RAs are not the best option when surgery is to be performed shortly since they induce platelet generation in the long term. However, their use could be considered to maintain suitable platelet counts after surgery, especially after complex procedures.

Finally, when surgery is going to be planned, the time spent for an agent to achieve a sufficient platelet count increase must be considered when setting the date of the procedure. There is currently no agent that should definitely be chosen ahead of others for presurgical preparation. The same therapies that are suitable for first- and second-line treatment can be used for presurgical preparation with the same hierarchy [66]. Table 5 summarizes the recommended platelet counts according to surgical risk as well as the guidelines for proceeding with urgent or planned surgical procedures.

### 4.4. Suspension of Treatment with TPO-RA

The long-term use of TPO-RA has allowed specialists to report long-lasting responses. This finding, together with the good safety profile associated with these drugs [67,68], has prompted their continuous use. Another argument to support this measure is the drop in platelet counts to pre-treatment values as early as 2 weeks after treatment suspension, which has been occasionally described [69]. Nevertheless, cases of long-term remission after treatment withdrawal (so-called sustained remission off-treatment (SROT)) have also been reported [37,38,70,71], which may be due to immunomodulatory actions performed by this therapeutic group [38]. This last observation encouraged some practitioners to reduce progressively the TPO-RA dose, and finally to suspend treatment, provided that a drop in the platelet count was not detected. This procedure not only saves costs but also reduces the risk of TPO-RA-associated adverse events [37]. Normally, candidates for achieving SROT after progressive dose reduction followed by suspension are those who present with stable platelet counts (50–100 × 10^9^/L) during a 4–6-month period on TPO-RA treatment, regardless of disease stage [37,38,71]. Patients must be properly informed about this therapeutic option for them to decide after balancing risks and benefits. Several protocols to reduce dosage and suspend treatment have been proposed [72,73,74,75]. Ours is detailed in Appendix A, where the profiles of SROT candidate and non-candidate patients are also described [73,74,76].

## 5. Primary ITP in Selected Patient Populations

### 5.1. Pediatric Patients

Primary ITP is usually self-limiting in children. The highest incidence is reported in 2–8 y.o. patients and a history of a triggering infectious episode is not an infrequent occurrence. The trend to spontaneous remission is observed even after 2 years of evolution. The diagnostic approach is similar to that in adults. Although most pediatric patients with newly diagnosed primary ITP do not present with relevant bleeding symptoms and do not require treatment, it is mandatory that parents and children be aware of the risks associated with a severe or potentially fatal hemorrhage.

Hospitalization is recommended for pediatric patients with active hemorrhage, bleeding risk factors or platelet counts ≤ 20 × 10^9^/L. In order to make therapeutic decisions, platelet counts should not be the only factor taken into account. Other variables such as mucocutaneous symptoms, the type of active hemorrhage and bleeding risk factors such as other hemostatic disorders and anticoagulant or antiplatelet drugs, should also be considered on a case-by-case basis. The aim of the treatment should focus on the control of clinically relevant hemorrhages rather than the platelet count recovery. First-line treatments are either corticosteroids, such as prednisone (oral) or methylprednisolone (i.v.), or high-dose IVIg. In the event of no response to the first chosen agent, the alternative one can be tried [77,78]. TPO-RA can be used as a second-line option [79,80]. Failure of the first and second treatment lines should prompt not only bone marrow examination but also the consideration of other drugs such as mycophenolate mofetil or rituximab, even though the experience with these agents is limited in children. Splenectomy may be an option in scenarios of life-threatening hemorrhage. Finally, dapsone has been shown to be effective and safe in pediatric ITP patients refractory to steroids and may be another third-line option [81]. Table 6 provides details about these therapeutic guidelines.

### 5.2. Elderly Patients

The incidence of primary ITP goes up to 9 per 100,000 individuals per year in >75 y.o. patients [1]. Nevertheless, the fact that some comorbidities causing thrombocytopenia can lead to an inaccurate diagnosis due to “ITP imitation” must be kept in mind. Furthermore, the incidence of these entities, such as megaloblastic anemias, myelodysplastic syndromes (MDS) or acute leukemias, increases with age. For this reason, differential diagnosis is particularly important. When reasonable doubts arise, bone marrow analysis, including cytogenetic and flow cytometry approaches, is recommended.

Elderly primary ITP patients are at higher risk of bleeding, thromboembolism and infection, and they frequently require antiplatelet and anticoagulant therapies. Platelet counts are the main determinants of bleeding risk and should be maintained at values >30 × 10^9^/L in >75 y.o. patients, as well as in those >60 years with concomitant bleeding risk factors [82,83]. TPO-RAs, IVIg and vinca alkaloids can be considered when a rapid platelet count increase is required [82].

The therapeutic attitude with elderly ITP patients with no active bleeding consists of the use of corticosteroids for first-line treatment, still at lower doses (prednisone at 0.5 mg/kg/day) and for shorter periods than those used with younger patients [1,82]. IVIg are indicated in the event of severe thrombocytopenia only (<10 × 10^9^/L) or with high bleeding risk [83]. According to the patient’s comorbidities, dexamethasone at standard doses may be an option. The choice of the second-line treatment should be made on an individual basis, and the patient should participate actively [1]. The good safety/efficacy profile of TPO-RAs in elderly patients makes them the main second-line therapeutic option [1,84]. Furthermore, their sustained response rates seem to be comparable to those observed with TPO-RAs in other adult populations [71,85,86]. Nevertheless, it must be remarked that the risk of thromboembolism associated with these drugs is higher in the elderly since the concomitant presence of several other thromboembolic risk factors is not uncommon [87]. An alternative option for patients at high thromboembolic risk can be fostamatinib [88,89]. Rituximab may also be considered, although long-term remissions are scarce, and more associated toxicities have been reported [1,83]. Finally, immunosuppressants or immunomodulators such as mycophenolate mofetil, cyclosporine, azathioprine, danazol or dapsone may be a valid option for those elderly patients presenting with moderate symptoms since the safety/efficacy profile of these agents is well-known. Nevertheless, many of these drugs require several months to achieve the intended effect [1,82]. Splenectomy is not recommended in the elderly, except in isolated cases of multirefractory patients, because the procedure is less effective and triggers more bleeding and infectious complications than in other populations. Table 6 provides details regarding the treatment of primary ITP in the elderly.

### 5.3. Pregnant Patients

When primary ITP is suspected in a pregnant woman, other pregnancy-related causes of thrombocytopenia should be ruled out. In fact, although thrombocytopenia is the second most frequently occurring hematologic disorder in pregnancy, around 80% of cases are of gestational origin. The hallmark of these is a progressive decrease in platelet counts, starting in the mid-second trimester and persisting in the third [90]. The procedure to diagnose primary ITP in pregnancy requires assessment of blood pressure, urine proteins, hemostatic status and antiphospholipid and antinuclear antibodies (ANAs) [91].

Severe complications are not frequently found in pregnant women with primary ITP, and neonatal incidence of thrombocytopenia or bleeding events is low. Particularly risky scenarios would be those of patients unable to maintain stable platelet counts > 30 × 10^9^/L with standard treatments, or patients with a history of previous pregnancies with severe neonatal thrombocytopenia. Recommended platelet counts to undergo vaginal delivery are >50 × 10^9^/L. This value goes up to >70 × 10^9^/L in the event that cesarean delivery is required or epidural anesthesia is going to be used. The choice of type of labor will be made according to obstetric criteria only [92].

Pregnant women with platelets > 30 × 10^9^/L do not systematically require treatment. With lower values, the first-line options are glucocorticoids and IVIg. Starting with prednisone is recommended. This should be used at doses of 10–20 mg/day since these are the lowest ones enough to achieve platelet counts in the range of 20–30 × 10^9^/L. Dexamethasone should not be used because it may induce adverse events for the fetus, such as oligohydramnios. IVIg has to be administered only in the event of side effects associated with steroids, severe hemorrhage or requirement for particularly rapid platelet count recovery, especially when delivery is close in time [7]. The usefulness of TPO-RAs as a second-line option has not been established yet, since enough clinical evidence is lacking (only isolated cases and one case series have been reported [93,94]). The data sheets for these drugs do not include this indication. The use of TPO-RAs during pregnancy should be considered only if the potential benefit to the mother justifies the potential risk to the fetus [93]. Furthermore, any decision concerning this medication should be made in accordance with the patient’s wishes, once she has been properly informed. If TPO-RAs are finally chosen, it is advisable to avoid them in the first trimester. Rituximab does not seem to be teratogenic. However, it has been associated with prolonged B-cell lymphocytopenia and the requirement to delay vaccination in neonates exposed in utero. For this reason, this agent should not be used within at least 6 months of planned conception [95]. Fostamatinib has been associated with fetal mortality in animal models [96].

Azathioprine and cyclosporin could be used without teratogenic risk. Nevertheless, the high rate of preterm birth and intrauterine growth retardation, which is associated with these medications, whose cause is not well-known, has to be kept in mind. Finally, data regarding the safety/efficacy of splenectomy in pregnant patients are limited. If the procedure is finally chosen, it should be performed during the second trimester, keeping in mind that an associated risk of neonatal thrombocytopenia exists [1].

After labor, platelet counts must be assessed in the neonate. If these are <100 × 10^9^/L, they should be monitored daily. With values < 50 × 10^9^/L, cranial ultrasound should be performed, and if hemorrhage was detected, IVIg and steroids, preferably prednisone, should be administered at minimal doses and for a short period of time, pursuing a platelet count target of >100 × 10^9^/L. There is little evidence about the use of platelet transfusion in this situation, but it could be useful in cases with no clinical improvement. The dose and frequency of transfusion should be adjusted according to platelet count and clinical evolution. In those neonates presenting with other bleeding locations or platelet counts < 30 × 10^9^/L, one unique dose of IVIg is recommended in order to achieve rapid response. Finally, those neonates with thrombocytopenia lasting beyond 3 weeks from birth should quit breastfeeding [1]. Table 6 summarizes the most relevant topics regarding the management of primary ITP in pregnant women and neonates.

## 6. Secondary ITP

Secondary forms of ITP account for 9–20% of all ITP cases in adults. This rate increases with age [97,98]. Those pathologies able to induce immune tolerance disorders leading to secondary ITP are varied. Systemic lupus erythematosus (SLE) is the most commonly found entity [99,100,101,102,103,104,105,106,107,108,109,110,111,112,113,114,115] (Table 7). Thrombocytopenias secondary to drug use are particular conditions [116,117]. Indeed, treatment with the causal agent must be immediately suspended. When this is heparin, another anticoagulant should be started, preferably i.v. administered thrombin direct inhibitors. After platelet count recovery, these can be substituted by coumarins, starting at low doses [118]. Direct oral anticoagulants might be another option, although there is not enough evidence to recommend them specifically yet. Finally, vaccines including aluminum as adjuvant have been reported to induce ITP occasionally. The severity of symptoms can vary, but there is a high responsiveness to IVIg even in severe cases [119].

The first-line treatment is similar in most cases of primary and secondary ITP, namely, glucocorticoids and/or IVIg. However, when choosing the second-line option, one has to consider seriously the underlying disease when managing secondary ITP. For instance, the benefit:risk ratio regarding TPO-RA use or splenectomy should be weighed up in cases of ITP secondary to SLE or antiphospholipid syndrome. Rituximab may be considered in the context of common variable immunodeficiency (CVID). The guidelines to treat secondary ITP are summarized in Table 7.

## 7. Primary ITP and Thrombosis

### 7.1. Pathophysiology, Risk Associated with the Treatment of Primary ITP

Patients with primary ITP are at twice the risk of venous or arterial thrombosis compared with the normal population, even when platelet counts are markedly low [120]. The origin is multifactorial, with causal roles played by the classical thromboembolic risk factors and the therapies that are being administered to treat thrombocytopenia [121]. On the one hand, patients with primary ITP have higher circulating levels of neutrophil extracellular traps (NETs), E-selectin, plasminogen activator inhibitor-1 (PAI-1) and microparticles rich in phosphatidylserine and tissue factor (TF), as well as hyperreactive immature platelets, within a proinflammatory scenario that also promotes coagulation, occasionally boosted by lupus anticoagulant and/or anticardiolipin or anti-β2-glycoprotein-I antibodies [122,123,124]. On the other hand, most primary ITP treatments induce some degree of thrombotic risk. Corticosteroids could increase the expression of TF and factor VIII, reduce that of thrombomodulin and promote cell adhesion via von Willebrand factor. Occasionally, IVIg could trigger thromboembolic venous events in patients with concomitant risk factors and arterial events in patients of advanced age and/or with atherosclerosis. Platelets of TPO-RA-treated patients tend to present apoptotic patterns leading to expression of phosphatidylserine on the cell surface, thus promoting the assembly of the prothrombinase complex. Finally, splenectomy may also promote thrombosis, either portal or systemic [72,124].

### 7.2. Antiplatelet and Anticoagulant Treatments in the Context of Primary ITP

It must be recalled that thrombocytopenia is predictive of a poor prognosis in patients with acute coronary syndromes. In order to minimize the bleeding risk associated with the use of antiplatelet agents in patients with thrombocytopenia, non-steroidal anti-inflammatory drugs and inhibitors of GPIIb/IIIa should be avoided, proton pump inhibitors should be administered, aspirin should be given at low dose, the prolonged use of triple antithrombotic therapy should be avoided and, in those patients undergoing stent placement, double antiplatelet therapy should be limited to one month after the procedure. Treatment should be decided on an individual basis and should be influenced by the thrombotic risk and the hemorrhagic history of each patient. Aspirin could be used in cases of acute arterial events provided that platelet counts are >10 × 10^9^/L, while the double antiplatelet treatment should be restricted to patients with counts >30 × 10^9^/L [125].

There are no studies designed to evaluate the safety and efficacy of anticoagulant treatment in primary ITP patients. Nevertheless, the administration of therapeutic doses of anticoagulants to patients with platelet counts > 50 × 10^9^/L is generally accepted. Bleeding risk increases when counts are <50 × 10^9^/L. In such cases, the options would be suspending anticoagulation or reducing the anticoagulant drug dose. In the event of total contraindication for anticoagulation, a vena cava filter could be placed, provided that the thrombus is below the placement area.

In those patients with a history of thromboembolism, glucocorticoids and fostamatinib would be the first-line and second-line options, respectively. In the event that there is no response to fostamatinib and platelet counts must be maintained to avoid complications associated with anticoagulation or antiplatelet drugs, the use of TPO-RAs could be considered.

Table 8 summarizes the guidelines to follow in the management of patients with primary ITP and thromboembolism or thromboembolic risk.

## 8. ITP and COVID-19

SARS-CoV-2, like other viral agents, is able to induce ITP. In this case, the diagnosis of secondary ITP is also by exclusion.

The treatment of ITP in patients with COVID-19 and platelet counts < 20 × 10^9^/L and/or active bleeding should consist of prednisone at 0.5–1 mg/kg/day for no more than 2 weeks, followed by progressive dose reduction and, finally, suspension no later than 8 weeks from the start. Those patients with severe COVID-19 who are already on corticoids and present with platelet counts < 20 × 10^9^/L and/or active bleeding could be additionally administered IVIg, at a total dose of 2 g/kg. In the event that counts continue to be <20 × 10^9^/L and/or active bleeding persists, TPO-RA could be administered, although at the lowest possible dose. An alternative option may be fostamatinib, which could be beneficial not only for platelet count recovery but also for relieving COVID-19-triggered inflammatory processes [126]. Rituximab, and other immunosuppressants, should be avoided since these agents reduce the ability to produce antibodies [127,128,129].

Those patients with primary ITP in its chronic phase who are being well-controlled with their ITP treatment should not change their therapeutic regimen if they are infected by SARS-CoV-2. Thrombocytosis may be observed in TPO-RA-treated patients, although it is usually transient and non-severe. In other cases, thrombocytosis occurs very rarely. Nevertheless, close monitoring is advisable [130,131,132,133]. If the infection leads to a relapse of the thrombocytopenia, patients should be administered IVIg if the drop in platelet counts is severe, and platelets should be transfused in the event of bleeding. In those patients who are already being treated with a TPO-RA, an increase in the dose or the addition of another TPO-RA or fostamatinib could be proposed [130,134].

When those patients with primary ITP, who are also on anticoagulant treatment, are infected by SARS-CoV-2, even if COVID-19 symptoms are severe, they can continue using low-molecular-weight heparin (LMWH) at prophylactic dose provided that cell counts are >30 × 10^9^/L. Anticoagulation or antiplatelet agents can be used at therapeutic doses with counts > 50 × 10^9^/L [130].

Finally, it must be remarked that the risk of secondary ITP associated with SARS-CoV-2 vaccination is very low, in the range of that induced by other commercially available vaccines against other viral agents [135,136]. There is no contraindication against using COVID-19 vaccines in pregnant women or patients with preexisting ITP [137,138].

The most important notions concerning the treatment of ITP secondary to COVID-19 and managing SARS-CoV-2 infection in patients with primary ITP are summarized in Table 8.

## 9. Limitations

The field of primary ITP is a rapidly changing landscape. Many of the guidelines and recommendations are aimed to provide some guidance only. Large series and/or randomized prospective studies to compare therapeutic approaches or assess reliably the efficacy of treatments are lacking in the primary ITP scenario. This is a common problem across all guides and consensus documents addressing this disorder. Recommendations have thus not been graded because they are taken from expert opinions and non-comparative studies and are therefore supported by only a low level of evidence. On the other hand, the pathophysiology has not been addressed in depth, since the aim of this article was to provide physicians with an updated reference for their day-to-day practice. Finally, secondary ITP deserves an updated review focusing exclusively on this complex condition.

## 10. Conclusions

Primary ITP management remains a challenge. Its diagnosis is performed by exclusion and requires the involvement of experienced practitioners. There is now a consensus on disease categories, criteria used to define refractoriness and types of treatment response, and all these should be considered when managing patients with thrombocytopenia. Glucocorticoids and IVIg remain the cornerstones of first-line treatment. Regarding second-line treatment, TPO-RAs are usually the first choice. Fostamatinib is also a valid option and has been shown to be even better than the former for those patients with high thromboembolic risk. A variety of immunosuppressants, immunomodulators or cytostatic agents may be considered for multirefractory patients, after carefully weighing up their risks and benefits. The increasing choice of therapeutic options means that splenectomy is now only performed on a very limited set of patients. Treatments may be adjusted in specific subpopulations (pediatric, elderly, pregnant women) or in the presence of concomitant conditions (thrombosis, COVID-19). Primary ITP management is continuously evolving, and regular updates are necessary.

## Figures and Tables

**Table 1 jcm-12-06422-t001:** Primary ITP: general aspects, pathophysiology and diagnosis.

**General Aspects**
Primary ITP is defined as a platelet count < 100 × 10^9^/L, which is not justified by any known reason. The term secondary ITP is limited to those situations where a platelet count drop to values < 100 × 10^9^/L is caused by diagnosed diseases able to induce immune destruction of platelets.
Incidence of primary ITP is two to four cases per 100,000 individuals per year in adults and children.
Bleeding manifestations are the main symptoms associated with primary ITP. Higher thrombotic risk, fatigue (occasionally unrelated to platelet counts) and higher predisposition to infection can also be observed. Adverse events subsequent to administration of primary ITP therapies are often seen.
The development of new treatments has probably improved prognosis of primary ITP patients. However, the exact influence of these therapies on causes of either mortality or the mortality rate has not been established.
**Pathophysiology**
Causal mechanisms underlying primary ITP lead to an increase in platelet destruction or a decrease in platelet generation.
Platelet destruction is caused by autoantibodies, phagocytes, complement, apoptosis and clearance through Ashwell–Morel receptors of hepatocytes.
The lower rate of platelet production is caused by autoantibodies able to block TPO function and by increased apoptosis of megakaryocytes.
**Diagnosis**
Diagnosis of primary ITP is performed by excluding systematically other causes of thrombocytopenia and is based essentially on clinical history, physical examination, CBC and peripheral blood smear.
Peripheral blood smear examination is paramount for diagnosis.
Additional studies may be required and should be requested according to presentation and clinical course of the disease.
Assessment of antiplatelet autoantibodies is not routinely indicated, although it may be useful in complex cases.
The systematic analysis of bone marrow is not recommended except in the event of refractoriness to treatments or when another disease is suspected. In these cases, bone marrow examination should include aspiration and biopsy, immunophenotyping with flow cytometry, cytogenetics and molecular biology.

CBC, complete blood count; ITP, immune thrombocytopenia; TPO, thrombopoietin.

**Table 2 jcm-12-06422-t002:** Consensual definitions and concepts in the context of ITP.

**Primary ITP Categories According to the Phase of the Disease** [4]
Newly diagnosed ITP: within 3 months from diagnosis.
Persistent ITP: between 3 and 12 months from diagnosis.
Chronic ITP: lasting for more than 12 months from diagnosis.
Severe ITP: there are bleeding symptoms sufficient to mandate treatment, or new bleeding symptoms requiring additional therapeutic intervention with either an increased dose or a different platelet-enhancing agent.
**Refractory ITP** [4]
Two criteria must be met:
Failure of splenectomy or subsequent relapse.
Severe ITP or bleeding risk that in the opinion of the attending physician requires therapy.
**Type of response** [4]
Complete response: platelet count ≥ 100 × 10^9^/L and absence of bleeding.
Response: platelet count ≥ 30 × 10^9^/L and at least doubling of the baseline count, and absence of bleeding.
No response: platelet count < 30 × 10^9^/L or less than doubling of the baseline count, or bleeding.
Corticosteroid dependence: the ongoing need for corticosteroid administration at least for 2 months to maintain a platelet count ≥ 30 × 10^9^/L and/or to avoid bleeding.
**Type of response** [6]
Durable response: platelet count ≥ 30 × 10^9^/L and at least doubling baseline at 6 months.
Early response: platelet count ≥ 30 × 10^9^/L and at least doubling baseline at 1 week.
Initial response: platelet count ≥ 30 × 10^9^/L and at least doubling baseline at 1 month.
Maintained response in the absence of treatment: response after 6 months without treatment.

Texts excerpted from Rodeghiero et al., 2009 [4] and Neunert et al., 2019 [6]. ITP, immune thrombocytopenia.

**Table 3 jcm-12-06422-t003:** First-line and second-line treatment of primary ITP and management of refractory patients.

**First-Line Treatment**
Decision relies basically on bleeding symptoms and platelet counts (<20 × 10^9^/L).
First-line treatment is glucocorticoids (prednisone 0.5–1 mg/kg or dexamethasone 40 mg/day for 4 days).
Treatment should not last more than 8 weeks in the case of prednisone or more than three cycles in the case of dexamethasone.
ERR prednisone: 60–80%; SRR prednisone: 30–50%.
Potential side effects of glucocorticoids other than those linked to immunosuppression are ecchymosis, skin thinning and atrophy, acne, mild hirsutism, facial erythema, stria, impaired wound healing, thinning of hair, perioral dermatitis and adverse gastrointestinal effects.
IVIg are reserved for patients with severe hemorrhage or when steroids are contraindicated.
ERR IVIg: 75–92%; SRR IVIg: 30–55%.
In severe hemorrhage scenarios, combined treatment is suitable (IVIg, high-dose methylprednisolone, platelet transfusion; consider whether antifibrinolytics and/or TPO-RA are required).
Potential side effects of IVIg are chills, fever, flushing, flu-like muscle pains or joint pains, nausea, fatigue, rash, vomiting and very rarely, allergic reactions or anemia.
Anti-D immunoglobulins may be used, although they are not available world-wide. The single, i.v. dose, is 50–75 μg/kg, ERR is 80–90% and SRR at 60 days is 17%. A potential side effect is secondary hemolytic anemia in Rh(D) positive patients.
Hospitalization for at least 48–72 h is recommended for newly diagnosed patients with platelet counts < 20 × 10^9^/L.
**Second-Line Treatment**
The first choice should be TPO-RA (eltrombopag, romiplostim, avatrombopag) or fostamatinib.
TPO-RA exhibits a good safety profile, although its cost is high. The choice of one TPO-RA or another should be based on administration route, patient preference and potential future complications.
ERR eltrombopag: 70–80%; SRR eltrombopag: 10–30%; ERR romiplostim: 70–80%; SRR romiplostim: 10–30%; ERR avatrombopag: 65%; SRR avatrombopag: not known.
TPO-RA may increase the risk of venous thromboembolism. Other potential side effects are headache, tiredness, arthralgias, nausea and nasopharyngitis.
Fostamatinib is a SYK inhibitor able to reduce the anti-platelet activity of phagocytes.
ERR fostamatinib: 18–43%.
Responses to fostamatinib are observed early, and good results in multirefractory patients have been described.
Fostamatinib is particularly suitable as a first option for second-line treatment in patients with high thromboembolic risk.
Potential side effects of fostamatinib are blood pressure increase, liver toxicity, severe diarrhea, and infections.
Rituximab should be the secondary scenario in second-line options. The more used regimen consists of four doses of 375 mg/m^2^ each, administered on a weekly basis. Nevertheless, the same temporal pattern reducing each dose from 375 to 100 mg/m^2^ has been shown to have the same efficacy, while being possibly safer.
ERR rituximab: 60–80%; SRR rituximab: 20–30%.
Potential side effects of rituximab other than those linked to immunosuppression are infusion-related reactions (special attention has to be paid to fever, chills, shaking, dizziness, trouble breathing, itching or rash, lightheadedness or fainting), body aches, tiredness and nausea.
Splenectomy can be considered in chronic phases after at least one second-line treatment has failed.
ERR splenectomy: 80–90%; SRR splenectomy: 60–70%.
The laparoscopic procedure is preferred if splenectomy is finally decided.
Potential side effects of splenectomy other than infection risk are pancreatitis/fistula, atelectasis, bleeding or pulmonary embolism.
**Refractory Patients**
There is no clear recommendation about how those refractory patient treatments should be managed.
Combined therapies are usually more effective than monotherapy in refractory patients. Ideally, agents with different mechanisms of action should be combined. Rescues have been described using steroids concomitantly with rituximab or TPO-RA.
In the event of no response to one treatment, adding a new therapy concomitantly may be better than suspending the former and starting with the new one only.
Other diagnoses, such as drug-induced thrombocytopenia, myelodysplastic syndrome or hereditary thrombocytopenia, should be considered in multirefractory patients.
The use of immunosuppressants, immunomodulators or cytostatic agents can be considered. Nevertheless, their side effects, especially those linked to an increased infection risk, make it advisable to balance carefully the benefit:risk ratio.

ERR, expected response rate; ITP, immune thrombocytopenia; i.v., intravenous; IVIg, intravenous immunoglobulins; SRR, sustained response rate; TPO-RA, agonist of thrombopoietin receptor; SYK, spleen tyrosine kinase.

**Table 4 jcm-12-06422-t004:** Guidelines for the follow-up of non-hospitalized primary ITP patients.

**Newly Diagnosed Patients**
The frequency of CBC should be established according to individual features in order to guarantee stable platelet counts.
The patient has to be educated to recognize alarm signs early (hemorrhage; fatigue; pregnancy; start of anticoagulant or antiplatelet treatment; planned invasive procedures).
Disorders that may lead to erroneous primary ITP diagnosis have to be ruled out: autoimmune diseases, thyroid disease, hematologic disorders, immunodeficiencies.
**Patients with persistent or chronic ITP who are not being treated**
CBC on a 3–6 monthly basis.
The patient has to be educated to recognize alarm signs.
If still pending, continue proceeding with the differential diagnosis to rule out other autoimmune diseases.
Attention should be paid to complications associated with the use of previous therapies.
**Patients currently on treatment** (In all cases, the aforementioned actions to be taken in risk situations are applicable.)
***With corticoids***
Patients have to be informed about the more important side effects: hyperglycemia; hypertension; sleep disorder; state of mind disorder; osteoporosis; muscle weakness or atrophy; weight gain; infection; acne; skin stretch
Prophylaxis of osteoporosis with calcium and vitamin D is recommended if steroids are used for >4 weeks.
Infection prophylaxis as described is recommended.
***With IVIg***
CBC has to be performed on a weekly basis to assess efficacy and, accordingly, duration of therapy.
Patients have to be informed about the possibility of suffering cephalea (occasionally with meningismus) in the days following treatment administration.
***With TPO-RA***
CBC has to be performed on a weekly basis until maintenance dose is reached. Thereafter, CBC will be performed on a 4–8 weekly basis provided that platelet counts remain >50 × 10^9^/L.
Those patients with previous history of thrombembolism or with thrombembolic risk factors have to be informed about thromboembolic risk.
Peripheral blood smear has to be carefully studied in the event that CVC results suggest the onset of fibrosis.
Patients have to be informed about side effects they may experience, such as fatigue, headache, muscle and joint pain or cutaneous symptoms, explaining that they will not be severe.
Liver function has to be monitored if eltrombopag is the chosen TPO-RA.
***With fostamatinib***
CBC has to be performed on a monthly basis until a maintenance dose is reached. Thereafter, frequency has to be adapted to each individual requirement.
Blood pressure has to be controlled to monitor hypertension risk.
Liver enzymes have to be monitored each 4–8 weeks.
Patients have to be informed about possible side effects they may experience such as diarrhea or abdominal discomfort.
***With rituximab***
Serological control of HBV (anti-HBc, HBsAg) has to be performed.
Patients have to be warned about infection risk, especially when they are being administered rituximab concomitantly with corticoids.
Attention must be paid to neurologic signs consistent with progressive multifocal leukoencephalopathy in order to allow its early detection, since this disorder has been associated with this therapy, although still not frequently.
In the event of imminent vaccination (including SARS-CoV-2 vaccine), it is advisable to be aware that rituximab may influence efficacy. The vaccination calendar had to be adapted accordingly.
***Before/after splenectomy***
Before surgery, patients have to be vaccinated against encapsulated bacteria (*Neisseria meningitidis*, *Streptococcus pneumoniae*, *Haemophilus influenzae*). Patients must also be vaccinated against influenza before surgery and each year. MMR and varicella: two doses administered 4–8 weeks (preferably three months) apart from each other in subjects without evidence of immunity. Tdap: three doses in naive patient or one boost if previously vaccinated. New boost every 10 years. Serogroup B meningococcal vaccine has to be considered for those younger than 25 years. In the event that splenectomy is performed as an emergency procedure, vaccination should be performed afterwards
After surgery, proper thromboembolic prophylaxis should be initiated according to the patient’s characteristics. Close monitoring is recommended to anticipate thromboembolic complications.
Re-vaccination against the aforementioned pathogens should be performed according to established guidelines of each country. Meningococcal and pneumococcal vaccine boosts are recommended every 5 years.
Patients have to be warned about the risk of infection after the procedure and have to be educated for them to early detect symptoms coherent with this complication. They have to be explicitly told to go to the doctor if they are experiencing febrile episodes lasting more than 48 h.

Anti-HBc, antibody to HBV core antigen; CBC, complete blood count; HBsAg, hepatitis B surface antigen; HBV, hepatitis B virus; ITP, immune thrombocytopenia; IVIg, intravenous immunoglobulins; MMR, measles, mumps and rubella; Tdap, tetanus/diphtheria/pertussis; TPO-RA, agonist of thrombopoietin receptor.

**Table 5 jcm-12-06422-t005:** Topics of interest regarding surgery in patients with primary ITP.

**Recommended Preoperatory Platelet Count**			
Associated risk	Description	Procedures	Platelet count
Minor	Non-vital and exposed organs Easy identification and hemostasis in the event of bleeding Limited dissection	Tooth cleaning Simple tooth extractions Local dental anesthesia Broncho-alveolar lavage	≥20–30 × 10^9^/L
Moderate	Vital organs Difficult identification and hemostasis in the event of bleeding Profound and/or extensive dissection	Complex tooth extractions Bronchoscopy with transbronchial biopsy Digestive endoscopy/biopsy Minor surgery Cesarean delivery Lumbar puncture	≥50 × 10^9^/L
Major	Above-described scenarios when bleeding can be life-threatening or compromise surgery Surgeries associated with frequent bleeding	Epidural anesthesia Major surgery CNS and eye surgery (except cataract)	≥70 × 10^9^/L ≥80 × 10^9^/L ≥100 × 10^9^/L
**Management of emergency surgeries**			
Time to surgery	Therapeutic options (one or more)	Remarks	
<12–24 h	Dexamethasone, 40 mg/day × 4 days IVIg, 1 g/kg/day × 2 days Peri/intra-surgical platelet transfusion	Contact blood bank to arrange strategy and required resources	
1–7 days	Dexamethasone, 40 mg/day × 4 days IVIg, 1 g/kg/day × 2 days	Platelet transfusion is a valid option for those cases where no response to previous measures is observed	
**Management of scheduled surgeries**			
Time to surgery	Therapeutic options		
<2 weeks	Dexamethasone, 40 mg/day × 4 days IVIg, 1 g/kg/day × 2 days TPO-RA Eltrombopag, 50 mg/day Romiplostim, 3 μg/kg/week Avatrombopag, 20 mg/day		
4 weeks	Dexamethasone, 40 mg/day × 4 days Prednisone, 0.5–1 mg/kg/day TPO-RA Eltrombopag, 50 mg/day Romiplostim, 3 μg/kg/week Avatrombopag, 20 mg/day		

CNS, central nervous system; ITP, immune thrombocytopenia; IVIg, intravenous immunoglobulins; TPO-RA, agonist of thrombopoietin receptor.

**Table 6 jcm-12-06422-t006:** Management of primary ITP in pediatric patients, elderly patients and pregnant women.

**Pediatric Patients**
Therapeutic decisions should not rely on platelet counts only. The type of bleeding manifestations and hemorrhagic risk factors have also to be considered.
The aim of the treatment should prioritize the control of clinically relevant hemorrhages.
*First-line options*
Oral prednisone or i.v. methylprednisolone, 4 mg/kg/day (maximum dose 180 mg/day in three daily doses) during 4 days, 2 mg/kg for 3 days, then suspend.
High-dose IVIg, one single dose of 0.8–1 g/kg.
*Second-line options*
In persistent ITP
(If Rh+) i.v. anti-D Ig, one dose of 50–75 μg/kg, one-hour perfusion.
Methylprednisolone, i.v., 30 mg/kg/day for 3 days, 2 h perfusion.
Dexamethasone, oral, 0.6 mg/kg/day (one daily dose, 40 mg/day maximum dose) for 4 days each month.
In chronic ITP
TPO-RA (long-term treatment).
Romiplostim, s.c., one weekly dose, initial dose 1 μg/kg, weekly increases of 1 μg (10 μg maximum dose) until platelet counts ≥ 50 × 10^9^/L are reached.
Eltrombopag, oral daily dose of 25 mg (<6 years) or 50 mg (≥6 years). If platelet counts remain <50 × 10^9^/L after 2 weeks, increase daily dose in 12.5 mg (<6 years) or 25 mg (≥6 years). This pattern is repeated until platelet counts > 50 × 10^9^/L are reached, never using daily doses >75 mg.
*Third-line options*
Mycophenolate mofetil, 20–40 mg/kg/day orally, in two daily doses (response in 4–6 weeks).
Rituximab, currently under surveillance for suspicion of risk of progressive multifocal leukoencephalopathy; furthermore, risk of infection due to prolonged B-cell depletion. Infusion has to be closely monitored to anticipate acute immunoallergic reactions.
Splenectomy: - In ITP of new or persistent diagnosis, if there is a bleeding emergency that is life-threatening and does not respond to previous treatment. - In chronic ITP, if there is a life-threatening bleeding emergency. - Can be considered in patients >5 y.o. and >2 years evolution who are symptomatic and refractory to previous treatments, provided that ITP interferes the normal life development.
Dapsone has shown a good efficacy/safety profile in pediatric patients refractory to steroids
**Elderly patients**
Differential diagnosis is particularly important to reliably discard other entities and avoid wrong therapeutic approaches.
The aim of the treatment is to maintain platelet counts ≥ 30 × 10^9^/L in patients >75 y.o. (or in those >60 y.o. if there are concomitant bleeding risk factors), and improve QoL.
*When there is severe bleeding*
Hospitalization and immediate instauration of treatment.
*First-line options*
(General measures: local hemostasis, platelet and/or RBC transfusion, TXA, suspension of hemostatic medication).
IVIg, 0.4–0.5 g/kg/day during no more than 5 days (controlling hydration and renal function). Administer concomitantly with corticosteroids.
Corticosteroids.
Prednisone, but change to second-line in the event that doses >5 mg/day were required for >3 months to maintain the desired platelet count. Do not prolong treatment beyond 6–8 weeks.
Dexamethasone (avoid if possible; if chosen, avoid administering more than two to three cycles; these should not exceed 20 mg/day or 4 days).
*Additional options when rapid increases in platelet counts are required*
TPO-RA: romiplostim, eltrombopag, avatrombopag.
Vinca alcaloids: vinblastine, vincristine.
*When there is no bleeding*
*First-line options*
Corticosteroids.
Prednisone, but change to second-line in the event that doses >5 mg/day are required for >3 months to maintain the desired platelet count. Do not prolong treatment beyond 6–8 weeks
Dexamethasone (avoid if possible; if chosen, avoid administering more than two to three cycles; these should not exceed 20 mg/day or 4 days).
IVIg (only with severe thrombocytopenia [<10 × 10^9^/L] or when bleeding risk is unacceptable), 0.4–0.5 g/kg/day for no more than 5 days, controlling hydration and renal function, and being administered concomitantly with corticosteroids.
*Second-line options*
TPO-RA (first choice, ahead of the other second-line drugs).
Eltrombopag, oral daily dose of 25–75 mg.
Romiplostim, s.c., weekly dose of 1 μg/kg; if needed, increase dose progressively, never exceeding 10 μg/kg, until the target platelet count is reached. We suggest starting with 3 μg/kg/week to optimize time to response.
Avatrombopag, oral daily dose of 20–40 mg (dose adjustment with respect to other adult populations is not required).
Fostamatinib, start with two oral daily doses of 100 mg, increase to 150 mg if required to reach the target. Recommended option when there is high thromboembolic risk.
Rituximab, four doses of 100 or 375 mg/m^2^ for 4 consecutive weeks (long-term remissions are scarce, and toxicity is higher).
*Other options*
Immunosuppressants or immunomodulators (if moderate disease): mycophenolate mofetil, cyclosporin, azathioprine, danazol, dapsone (well-characterized profiles of safety/efficacy).
**Pregnant patients**
Before making therapeutic decisions, the differential diagnosis must be carefully assessed in order to rule out other causes of thrombocytopenia, especially those which are pregnancy-related.
Patients with platelet counts ≤ 20–30 × 10^9^/L require treatment. To undergo delivery, the recommended target for platelet count is >50 × 10^9^/L for vaginal and >70 × 10^9^/L for cesarean or if epidural anesthesia is going to be used.
*First-line options*
Prednisone, 10–20 mg/day, using the lowest possible dose that is enough to reach platelet counts in the range of 20–30 × 10^9^/L.
IVIg (daily dose of 1 g/kg for 2 days or daily dose of 0.4 g/kg for 5 days), in the event of prednisone-induced side effects, severe bleeding or requirement of rapid recovery of platelets to prepare for deliver.
*Other options*
Azathioprine, cyclosporin. If splenectomy is decided (data regarding safety/efficacy are limited, risk of neonatal Thrombocytopenia), the procedure should be performed in the second trimester.
Management of neonates will depend on their platelet count values. If these are <100 × 10^9^/L, repeat on a daily basis. If these are <50 × 10^9^/L, perform cranial ultrasound. If hemorrhage is detected, administer IVIg and steroids, pursuing a platelet count target of >100 × 10^9^/L. Although there is no evidence about what is the most suitable steroid, a short course of methylprednisolone could be a good option. If these are <30 × 10^9^/L or there are hemorrhagic symptoms, administer one single dose of IVIg (1 g/kg) to achieve rapid response.

Ig, immunoglobulins; ITP, immune thrombocytopenia; i.v., intravenous; IVIg, intravenous immunoglobulins; QoL, quality of life; s.c., subcutaneous; RBC, red blood cell; TPO-RA, agonist of thrombopoietin receptor; TXA, tranexamic acid; y.o., years old.

**Table 7 jcm-12-06422-t007:** Secondary ITP, causes and management.

Disruption of Immune Tolerance/Underlying Etiology	(%) *	Management
Central		
ALPS	1	Treat in case of lymphoproliferation or immune cytopenia (needed in 50% of cases due to onset of autoimmunity). Immunosuppressants or, in very selected cases, splenectomy, can be considered [99].
SLE	5	Treat when platelet counts are <20–30 × 10^9^/L. Corticoids are recommended for the first-line treatment, although early relapses are not infrequent. Recent recommendations suggest rituximab as a second-line option. In refractory cases, immunosuppressants (azathioprine, cyclosporine, cyclophosphamide, mycophenolate mofetil), splenectomy, TPO-RA and belimumab, alone or concomitantly with rituximab, have also been used [100,101].
Evans syndrome	2	The first-line treatment choice is corticosteroids. IVIg, rituximab, splenectomy and immunosuppressants have also been used. In cases associated with genetic abnormalities, therapies against the corresponding genetic target have been suggested [102].
PAPS	2	We should recall that, in clinical practice, patients with thrombocytopenia and antiphospholipid antibodies who do not meet the criteria for APS are frequently found. ITP associated with APS is treated similarly to SLE-associated ITP [103].
Differentiation		
CVID	1	Prednisone, 1 mg/kg for at least 3 weeks with subsequent dose decrease and final suspension can be used as a first-line choice. In case of an unsatisfactory response, IVIg, 1 g/kg, can be used. Both therapies could be concomitantly administered when rapid response is required. In refractory patients, rituximab, 375 mg/m2/week for 4 weeks, can be used. Splenectomy is recommended only when the first- and second-line treatments have failed. The risk of infection associated with splenectomy and immunosuppressants should always be borne in mind [104].
Lymphoproliferative syndromes/CLL	2	Corticosteroid therapy with predniso(lo)ne at an initial dosage of 1–2 mg/kg/day is the first-line treatment. The response to polyvalent immunoglobulins is not as good and should be reserved for emergency situations. TPO-RAs, although inconsistently effective, may be useful in this context. In the absence of concomitant CLL progression, rituximab monotherapy is a second-line option. Immunosuppressive therapies (eg, ciclosporin) may also be used. In the case of refractoriness or association with other features of CLL progression, treatment for CLL should be indicated according to national recommendations in each area [105,106,107].
ITP post-transplantation	1	The most frequently found profile corresponds to patients <3 months who have undergone umbilical cord cell transplantation. The therapeutic regimen is well-defined: first-line, IVIg 1 g/kg/day for 3 days, and monitoring weekly. In the event of an inadequate response, methylprednisolone can be used at 2 mg/kg/day for 14 days with subsequent dose decrease and suspension, not beyond 8 weeks from the start. TPO-RAs are valid options in the second-line treatment [115]. As a third-line choice, rituximab at 375 mg/m^2^/dose up to four doses (controlling IgG) can be used; bortezomib, mycophenolate mofetil or sirolimus may be options for multirefractory patients [108].
Peripheral immune response		
Viral infection	6–7	Frequent in pediatric viral infection. Described in association with CMV, EBV, VZV, ZIKV, HIV, HCV and SARS-CoV-2, as well as in post-vaccination periods. Usually, specific treatment is not required. If needed, IVIg should be used ahead of corticoids, as the latter could facilitate viral replication. The efficacy of antiviral therapy to accelerate cytopenia resolution has not been demonstrated [109]. TPO-RAs, especially avatrombopag, are indicated to manage HCV-induced thrombocytopenias [110]. The use of eltrombopag and romiplostim in patients with HIV-induced thrombocytopenia has been described as safe [111]. In the presence of SARS-CoV-2 management is similar to that described for primary ITP, although the use of TPO-RAs has to be carefully balanced due to the risk of thromboembolism and liver toxicity [112].
Helicobacter pylori	1	Although its association with thrombocytopenia has not been demonstrated beyond doubt, there are studies showing that the options to achieve platelet count recovery increase 14–15-fold after eradication therapy has been administered [113]. Thus, in these cases, the treatment to eradicate Helicobacter pylori (which, furthermore, is well-defined) could also be useful to overcome thrombocytopenia [114].
Drugs	n.c.	The web link “Platelets on the Web” is useful to review the drugs that have been associated with ITP [116]. Since these are numerous, the associated pathophysiological mechanisms are manifold. The incidence is one case/100,000 inhabitants/year, although this value is probably underestimated. On the other hand, there are vaccines that, although rarely, could also induce ITP, especially those including aluminum as adjuvant [119]. The treatment with the drug causing ITP should be immediately suspended. If the drug is heparin, another anticoagulant should be started, preferably i.v. administered thrombin direct inhibitors [118]. Since symptoms are usually resolved in the 2 days following drug suspension, there is no need for specific therap. In severe cases with associated bleeding, corticoids, IVIg or platelet transfusion have been used [117].

Modified from Cines et al. [100]. * Percentage with respect to the total amount of ITP (primary and secondary). ALPS, autoimmune lymphoproliferative syndrome; APS, anti-phospholipid syndrome; CLL, chronic lymphocytic leukemia; CMV, cytomegalovirus; CVID, common variable immune deficiency; EBV, Epstein–Barr virus; HCV, hepatitis C virus; HIV, human immunodeficiency virus; IgG, immunoglobulin G; ITP, immune thrombocytopenia; i.v., intravenous; IVIg, intravenous immunoglobulins; n.c., non-calculated; PAPS, primary anti-phospholipid syndrome; SLE, systemic lupus erythematosus; TPO-RA, agonist of thrombopoietin receptor; VZV, varicella zoster virus; ZIKV, Zika virus.

**Table 8 jcm-12-06422-t008:** Primary ITP in special scenarios: thrombosis, COVID-19.

**Primary ITP and Thrombosis**
The risk of thrombosis, either venous or arterial, is 2-fold higher in patients with primary ITP.
The origin is multifactorial, and there are many actors of primary hemostasis, coagulation and fibrinolysis playing a role. Some therapies for primary ITP also contribute to the increase in thromboembolic risk.
Thrombocytopenia is associated with a poorer prognosis in patients with acute coronary syndromes. Thus, platelet count recovery must be a priority target. Treatment must be individualized according to hemorrhagic history and thromboembolic risk. In an acute arterial episode with platelet counts > 10 × 10^9^/L aspirin could be used, while double antiplatelet treatment may be considered with platelet counts > 30 × 10^9^/L.
Anticoagulants can be used at full doses with platelet counts > 50 × 10^9^/L. With lower counts, the options are either dose reduction or suspension. In those situations where anticoagulation is contraindicated while immediate measures are required, a vena cava filter can be used or, with platelet counts < 10 × 10^9^/L, prophylactic platelet transfusion could be performed.
In patients with thromboembolic history, the preferred choices for first- and second-line treatment are glucocorticoids and fostamatinib, respectively. Only in the event of no response to the latter, if maintenance of the platelet count is required in order to continue administering antiplatelet or anticoagulant treatment safely, the use of TPO-RAs could be considered.
**Primary ITP and COVID-19**
The diagnosis of ITP in the context of COVID-19 is a diagnosis of exclusion.
Patients with platelet counts < 20 × 10^9^/L and/or active bleeding have to be treated with prednisone, 0.5–1 mg/kg/day for no more than 2 weeks, with progressive reduction and suspension not beyond 8 weeks from the start.
Those patients with severe COVID-19 who are already with corticoids and present with platelet counts < 20 × 10^9^/L and/or active bleeding could be additionally treated with IVIg, 2 g/kg total dosis. If counts < 20 × 10^9^/L and/or active bleeding persist, one TPO-RA could be administered, although, at the lowest possible dose, Fostamatinib may be one alternative option to TPO-RA.
Rituximab must be avoided, since the patient’s ability to produce antibodies would be compromised. For the same reason, other immunosuppressants should also be avoided whenever possible.
When patients with chronic primary ITP who are being well-controlled with treatment are infected by SARS-CoV-2, their therapeutic regimen should not be modified. In the event that the infection induces a relapse, IVIg should be administered in case of severe thrombocytopenia and, if bleeding occurred, a platelet transfusion could be performed. Those patients who are already under treatment with TPO-RA could consider either a dose increase or the addition of another TPO-RA or fostamatinib.
If patients with primary ITP who are receiving anticoagulant/antiplatelet treatment are infected by SARS-CoV-2 and present with severe symptoms:
If they are on LMWH, they can continue treatment at full dose provided that platelet counts are >30 × 10^9^/L
If they are on other anticoagulant or antiplatelet agents, treatment at full dose could be administered with platelet counts > 50 × 10^9^/L.
The risk of secondary ITP subsequent to SARS-CoV-2 vaccination is not higher than that induced by another antiviral vaccines; SARS-CoV-2 vaccine is not contraindicated in pregnant women or patients with history of ITP.

ITP, immune thrombocytopenia; IVIg, intravenous immunoglobulins; LMWH, low molecular weight heparin; TPO-RA, agonist of thrombopoietin receptor.

## Data Availability

This article did not report any data.

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
