# Peer review of "Recommendations for the Clinical Approach to Immune Thrombocytopenia: Spanish ITP Working Group (GEPTI)"

_jcm, 2023, doi:10.3390/jcm12206422_

Round 1

Reviewer 1 Report

The paper is well written and represents a significant contribution to the overall treatment of ITP in the era of new drugs. However, minor clarifications and corrections are necessary.

Title and abstract

You stated that recomendations are for patients with primary ITP. However You have a full chapter about treatment of secondary ITP.

Table 3. First-line and second-line treatment of primary ITP, and management of refractory patients.

For the second line therapy option it will be fair to present all drugs/therapy on the same way, eg EER, SRR, side efects and price

Second line therapy

The findings observed in studies with TPO-RAs, fostamatinib and rituximab suggest that the first two are the more effective and less toxic therapeutic options to be used as second-line treatment of primary ITP [6,7].

The two references listed, the ICR and the ASH 2019 giedlines, do not support your claim. What is the level of evidence for the same_ Although I agree with the idea behind the sentence, I think we should clarify on the basis of which studies the same dinette is based on.

Furthermore, many patients will be able to suspend other treatments,even TPO-RAs themselves, without leading to a new drop in platelet counts.

It would be useful to state exactly how much. Since „many“ it is quite uncertain for me

In the absence of response to a TPO-RA, switching to another one or to fostamatinib is recommended.

It wolud be useful to state percentig of patients who will repont to fosamatinig after TPO-RA failure

If TPO-RA refractoriness is definitely confirmed, the use of other immunosuppressants such as mycophenolate mofetil, azathioprine or low-dose steroids can be considered [23].

You do nor recomend rituximab usage in this situation?

Furthermore, some studies have shown encouraging results regarding long-term efficacy of fosamatinib

It would be useful to state data about long-term efficacy and stability of response

One again as for table 3 for the second line therapy option it will be fair to present all drugs/therapy on the same way, eg EER, SRR, side efects and price

Pregnant patients

Administration of drugs during pregnancy is a very sensitive and important issue. Therefore, it seems to me that authors must be clearer and more precise.

The usefulness of TPO-RAs as second-line option has not been established yet, since enough

clinical evidence is lacking (only isolated cases and one case series have been reported

[79,80]). The data sheets of these drugs do not include this indication, and any decision

concerning this medication should be made in accordance with the patientʹs wishes, once

she has been properly informed.

Having in minde that Bussel et al papaer included 186 women exposed to romiplostim author should report data of pregnancy outcas. Bussel et al stated that romiplostim should be used during pregnancy as treatment for maternal ITP only if the potential benefit to the mother justifies the potential risk to the fetus.

Azathioprine and cyclosporin can be used without teratogenic risk.

But with high rate of preterm birth and IGR.

Secondary ITP

Lymphoproliferative syndromes/CLL

Response to corticoids and IVIg is worse than in primary ITP. However, good responses to rituximab have been reported In refractory cases, TPO-RA may be used, and splenectomy could be considered In severe and multirefractory cases, treatment specific for CLL should be administered in the absence of other criteria of treatment to control thrombocytopenia [85]

The author's recommendations are in contrast with the currently valid recommendations for the specified group of diseases. It is necessary to state the level of evidence and incorporate new references.

ITP and COVID-19

Those patients with primary ITP in its chronic phase who are being well controlled with their ITP treatment, should not change their therapeutic regimen if they are infected by SARS-CoV-2.

Several authors pointed out that during the COVID 19 infection of patients with ITP on TPO RA therapy, extreme thrombocytosis can occur.

Pantic N, Suvajdzic-Vukovic N, Virijevic M, Pravdic Z, Sabljic N, Adzic-Vukicevic T, Mitrovic M. Coronavirus disease 2019 in patients with chronic immune thrombocytopenia on thrombopoietin receptor agonists: new perspectives and old challenges. Blood Coagul Fibrinolysis 2022;33(1):51-55

Wang Z, Cheng X, Wang N, Meng J, Ma J, Chen Z, Wu R.  Transient increase in platelet counts associated with COVID-19 infection during TPO-RA as the second-line treatment in children with ITP. Br J Haematol. 2023 Aug 23. doi: 10.1111/bjh.19040.

Reviewer 2 Report

As the authors stated, there is a number of guidelines and several recommendations for ITP in adults and children. Although it is demanding to write a review article on this topic, in order to significantly contribute to the fiels, these guidelines should be summarize according to the level of evidence. 

The role of vitamin D in ITP, like Helicobacter pylori eradication, is not mentioned and deserves comments too. 

The Introduction should be structured, not mixed clinical finding/definitions/diagnosis irregularly. Thrombotic risks are not even mentioned and should be added in the short ITP description.

Table 4 - Why is astenia alarm sign? Why not muscle weakness but immediately atrophy?

Table 5 - What are headeache signs?

Please reconsider splenectomy as the second-line treatment in particular situations, not generally.

Pediatric ITP - Please explain "the nature of mucocutaneous symptoms". Do not repeat that Plt count is not the indication for the treatment.

Explain bleeding risk factors in children.

Pregnancy - What do you mean "to avoid fetal risk" - which kind of risk?

Describe what steroid and regimen shoud be applied in te case of neonatal intracranial hemorrage. What about Plt transfusion in these cases?

Table 7 - Name the most frequent ITP-related vaccine. Add the link to "Platelets on the Web"

? Terms: prednisona, hyperglucemia, a posteriori, Pneumocistiis, neonatal incidences, solid diagnosis...

Reviewer 3 Report

This is an extensive review/ guideline. 

However, a few suggestions are placed to further improve the manuscript.

Comment 1: First line treatment: Like corticosteroid and IVIg, the role of anti-D Immunoglobulin must be mentioned in the text/ table (please add and comment these references about antiD Ig: Mishra K, Kumar S, Singh K, et al. Real-world experience of anti-D immunoglobulin in immune thrombocytopenia. Ann Hematol. 2022;101(6):1173-1179. doi: 10.1007/s00277-022-04829-4.

Sandal R, Mishra K, Jandial A, Sahu KK, Siddiqui AD. Update on diagnosis and treatment of Immune thrombocytopenia. Expert Review of Clinical Pharmacology. 2021 Mar;14(5):553-568. DOI: 10.1080/17512433.2021.1903315)

Comment 2: Table 4: It is advisable to mention the revaccination schedule with doses rather than mentioning as per guidelines

Comment 3: Pneumocistiis carinii may be changed to Pneumocystis jirovecii

Comment 4: Rituximab: please mention the better efficacy observed in patients having ANA positivity (please add and comment these references about Rituximab and specially rituximab with ANA positivity: Mishra K, Kumar S, Jandial A, Sahu KK, Sandal R, Ahuja A, et al. Real-world Experience of Rituximab in Immune Thrombocytopenia. Indian J Hematol Blood Transfus. 2021 Jul;37(3):404-413. doi: 10.1007/s12288-020-01351-3.)

Comment 5: Splenectomy: many reports on long term outcome of splenectomy have not reported any serious concerns about the infection or thrombosis risks and it is particularly important in the present era with wide availability of vaccine and thromboembolic prophylaxis; this aspect of splenectomy must also be mentioned. (please add and comment these references about long term safety profile of splenectomy in ITP: Mishra K, Kumar S, Sandal R, Jandial A, Sahu KK, Singh K, et al. Safety and efficacy of splenectomy in immune thrombocytopenia. Am J Blood Res. 2021 Aug 15;11(4):361-372.)

Comment 6: Table 6: dapsone has very good efficacy and safety in steroid refractory ITP of pediatric age group, hence advisable to be mentioned as one of the options (3rd line) (please add and comment these references about dapsone: Khera S, Pramanik SK, Yanamandra U, Mishra K, Kapoor R, Das S. Dapsone: An Old but Effective Therapy in Pediatric Refractory Immune Thrombocytopenia. Indian J Hematol Blood Transfus. 2020 Oct;36(4):690-694. doi: 10.1007/s12288-020-01286-9.)

Comment 7: does iron deficiency anaemia imitate ITP?

Comment 8: Some references may be added as advised above or in the manuscript.

Comment 9: Was there any treatment offered to the patients for cytopenia before or after bone marrow study ?

Comment 10: At the end of Introduction Section indicate the aim of this review.

Comment 11: As there are many existing reviews on this topic and recently two international guidelines have been published. Authors should explain what their guideline contributes.

Comments 12: The authors should also discuss the clinical trials of combination frontline treatment for ITP (steroids and rituximab, eltrombopag or Mycophenolate)

There are multiple spelling/ grammatical errors

Round 2

Reviewer 3 Report

Very well modified